# Stromal COL11A1: Mechanisms of Stroma-Driven Multidrug Resistance in Breast Cancer and Biomarker Potential

**DOI:** 10.3390/biomedicines13122905

**Published:** 2025-11-27

**Authors:** Andreea Onofrei (Popa), Felicia Mihailuta, Daniela Mihalache, Cristina Chelmu Vodă, Sanda Jurja, Sorin Deacu, Mihaela Cezarina Mehedinți

**Affiliations:** 1Faculty of Medicine and Pharmacy, “Dunarea de Jos” University, 800008 Galati, Romania; andreea.onofrei@ugal.ro (A.O.); felicia.mihailuta@yahoo.com (F.M.); mihaela.mehedinti@ugal.ro (M.C.M.); 2Research Centre in the Medical-Pharmaceutical Field, “Dunarea de Jos” University, 800010 Galati, Romania; 3Faculty of Medicine and Pharmacy, Ovidius University, 900470 Constanța, Romania; jurjasanda@yahoo.com (S.J.); deacu.sorin@gmail.com (S.D.)

**Keywords:** breast cancer, chemotherapy, COL11A1, hormone therapy, immunotherapy

## Abstract

**Background/Objectives:** Therapeutic resistance remains a major obstacle in breast cancer management, particularly among estrogen receptor-positive (ERα^+^) tumors that initially respond to endocrine therapy such as tamoxifen. Type XI collagen (COL11A1), a minor fibrillar collagen secreted by cancer-associated fibroblasts, has recently emerged as a stromal biomarker linked to tumor progression, immune modulation, and poor prognosis in several solid malignancies. **Methods:** We conducted a narrative review of the literature indexed in PubMed, Scopus, and Web of Science between 2011 and 2025, including original research, reviews, and clinical studies addressing COL11A1 expression and function in breast cancer. Mechanistic studies in other cancer types (ovarian, pancreatic, lung) were also evaluated when relevant to breast cancer biology. **Results:** Across multiple cancer types, COL11A1 overexpression correlates with stromal remodeling, epithelial–mesenchymal transition, and resistance to both hormone therapy and chemotherapy. In breast cancer, emerging data suggest a potential prognostic role and possible involvement in shaping the immune microenvironment. Nevertheless, most evidence derives from retrospective or preclinical studies, and clinical validation remains limited. **Conclusions:** COL11A1 represents a promising, though still exploratory, biomarker of therapeutic resistance and immune modulation in breast cancer. Future prospective and subtype-specific studies are needed to clarify its diagnostic and therapeutic value and to determine whether its inclusion in immunohistochemical panels could enhance patient stratification and guide personalized treatment.

## 1. Introduction

Breast cancer remains the most common malignancy among women worldwide and a leading cause of cancer related mortality.

According to GLOBOCAN 2022, over 666,000 deaths were attributed to breast cancer globally, with notable regional variations in incidence and mortality across Asia and Europe [1]. Despite advances in screening and therapy, many patients are still diagnosed at advanced stages due to the often insidious progression of the disease [2].

Standard treatments including surgery, chemotherapy, endocrine therapy, and targeted therapies have improved survival. However, resistance to hormone therapy and chemotherapy continues to pose a major clinical challenge. This highlights the urgent need for novel prognostic biomarkers and innovative therapeutic strategies [2].

Immunotherapy has emerged as a promising option for selected subgroups of breast cancer patients, particularly those with triple-negative breast cancer (TNBC). Yet, predictive biomarkers of response remain limited, and only a subset of patients benefit from immune checkpoint inhibitors. In this context, type XI collagen, specifically its α1 chain (COL11A1), has gained increasing attention [2,3].

COL11A1 is a minor fibrillar collagen normally expressed in tissues such as cartilage, the vitreous body, and mesenchymal cells, but it is aberrantly overexpressed in the tumor stroma of several cancers, including breast cancer, see Figure 1. Its expression has been linked to tumor angiogenesis, invasion, immune cell infiltration, therapy resistance, and poor prognosis [4].

Mechanistically, COL11A1 is predominantly produced by cancer-associated fibroblasts (CAFs) within the tumor microenvironment. It contributes to stromal remodeling, epithelial–mesenchymal transition, a process by which epithelial cells acquire migratory and invasive properties, and metastatic dissemination. While much of the mechanistic understanding of COL11A1 comes from studies in ovarian, lung, and pancreatic cancers, emerging evidence suggests a similar role in breast cancer progression and therapy resistance. Nonetheless, most data remain preclinical or retrospective, and its translational relevance, particularly in the context of immunotherapy, is still under investigation [5].

This review aims to synthesize current knowledge on COL11A1 in breast cancer, focusing on its involvement in therapeutic resistance, tumor immune interactions, and its potential as a diagnostic, prognostic, or therapeutic biomarker.

## 2. Materials and Methods

To carry out this narrative review, we conducted a comprehensive search of the scientific literature to synthesize existing data on the role of COL11A1 in the pathogenesis and progression of breast cancer. Searches were performed in PubMed, Scopus, and Web of Science, between 2011 and 2025. The following keywords were used: breast cancer, chemotherapy, COL11A1, hormone therapy, and immunotherapy.

We included original research articles, review articles, and clinical studies on breast cancer, as well as non-breast cancer studies (ovarian, pancreatic, lung) that provided mechanistic insights potentially relevant to breast cancer. Exclusion criteria were non-English publications, case reports, and conference abstracts without full text.

Articles were initially screened based on title and abstract, followed by full text review for eligibility. After this process, a total of 72 studies met the inclusion criteria. Table 1 summarizes the relevant clinical studies included in this review.

In writing this narrative review, artificial intelligence tools such as ChatGPT-5.1 were used as support for: superficial text editing such as grammar and formatting.

## 3. Breast Cancer Subtypes and Stromal Drivers of Therapeutic Resistance

Breast cancer is the most common malignancy among women worldwide, with over 2.3 million new cases diagnosed annually since 2020. Approximately 30% of patients develop therapeutic resistance during the disease course, which has become the leading cause of death [11,12,13].

Although molecular subtyping (Luminal A/B, HER2-enriched, basal-like, etc.) has advanced personalized treatment, the challenge of drug resistance remains unresolved. For instance, triple-negative breast cancer (TNBC), which lacks well-defined therapeutic targets, has a chemotherapy resistance rate exceeding 50%. Additionally, about 60% of HER2-positive breast cancer patients develop secondary resistance within 1–2 years of targeted therapy [14,15,16,17,18,19,20].

Existing biomarkers associated with drug resistance mostly focus on tumor cells themselves (e.g., ER/PR expression, HER2 amplification, BRCA mutations), while overlooking the regulatory role of the tumor microenvironment (TME) [12,13]. As core components of the TME, stromal cells and the extracellular matrix (ECM) they secrete play a pivotal role in drug resistance. However, there are very few studies on stromal biomarkers to date, resulting in a lack of effective predictive and interventional targets in clinical practice [21,22,23,24,25,26].

From the perspective of subtype characteristics, the drug resistance mechanisms of different breast cancer subtypes are closely related to the stroma. Endocrine resistance in luminal breast cancer is often accompanied by marked stromal fibrosis. “Cold tumor” subtypes, such as basal-like 1 (BL1) and mesenchymal (M) in TNBC, are insensitive to both chemotherapy and immunotherapy. This insensitivity is due to a thick stromal barrier and poor immune cell infiltration [14,15].

This suggests that targeting stromal components may emerge as a new direction to overcome drug resistance, and the role of COL11A1 — a stroma-specific molecule discovered in recent years — in regulating drug resistance urgently needs to be elucidated.

## 4. COL11A1 and Classical Immunohistochemical Markers

Compared to classical immunohistochemical markers such as estrogen receptor (ER), progesterone receptor (PR), and human epidermal growth factor receptor 2 (HER2) which are routinely used for molecular classification and therapeutic decision making in breast cancer, COL11A1 exhibits a distinct biological and pathological profile. The traditional markers ER, PR, and HER2 are expressed directly by tumor cells and have validated predictive value for response to endocrine and anti-HER2 therapies [26].

According to Qi Luo et al., a correlation exists between the immunohistochemical expression of COL11A1 and conventional markers (ER, PR, HER2), as well as with clinical parameters including primary tumor grade (T), number of affected lymph nodes (N), and the presence or absence of distant metastases (M). Their study concluded that coexpression of ER, PR, and COL11A1 does not significantly influence overall survival, whereas the combined expression of HER2 and COL11A1 correlates with poorer outcomes. Furthermore, COL11A1 expression is associated with unfavorable prognosis exclusively in patients with Nottingham histological grade 3 tumors [2].

In contrast to classical markers, COL11A1 is predominantly expressed in stromal cells particularly in cancer-associated fibroblasts (CAFs), rather than in tumor epithelial cells. This pattern reflects active remodeling of the tumor microenvironment rather than intrinsic tumor cell features. Consequently, COL11A1 may complement, rather than replace, traditional biomarkers by providing additional insights into tumor invasiveness, angiogenesis, and therapy resistance [5].

Tommaso Susini et al. demonstrated that, immunohistochemically, COL11A1 is more frequently expressed in the cytoplasm of fibroblasts within the peritumoral stroma, and less frequently in tumor epithelial cells. High expression is defined when ≥15% of stromal fibroblasts show cytoplasmic positivity for COL11A1, low expression corresponds to <15% positivity, while undetectable expression indicates complete absence of staining [27]. In the context of immunotherapy, PD-L1 remains the principal biomarker for selecting patients eligible for immune checkpoint inhibitor therapy [28].

However, unlike PD-L1, whose expression is heterogeneous and dynamic, COL11A1 may reflect a more stable marker of aggressive tumor behavior and tumor stroma immune interactions.

COL11A1 provides valuable information about the tumor microenvironment and may assist in: confirming stromal invasion in diagnostically uncertain cases (e.g., DCIS or ADH), assessing recurrence risk, and evaluating response to antistromal therapies [6].

In core biopsies, where identifying stromal invasion can be challenging, COL11A1 may serve as a useful adjunct marker. It should be interpreted in conjunction with classical markers such as p63, calponin, and ER/PR/HER2, rather than as a replacement, see Table 2 [29].

A study by Freire et al. found that COL11A1 demonstrates superior sensitivity in detecting microinfiltration in breast cancer compared to traditional diagnostic markers such as p63 and calponin, while also being easier to interpret histologically. The study analyzed 40 cases of breast carcinoma in situ, of which 21 exhibited tumor microinfiltration, identified through COL11A1 immunohistochemical expression [6].

In conclusion, unlike ER, PR, HER2, and PD-L1, COL11A1 is not yet clinically validated and remains an emerging biomarker. Further studies are required to establish its definitive prognostic and predictive value in breast cancer.

## 5. COL11A1 an Important Stromal Factor in Breast Cancer

### 5.1. COL11A1-Tumor Microenvironment

It has been observed that COL11A1 is overexpressed in the desmoplastic stroma of tumors predominantly formed by cancer-associated fibroblasts and is not expressed by fibroblasts associated with inflammatory diseases [7]. Studies have further demonstrated that CAFs can originate from multiple cellular sources, including adipocytes, bone marrow derived mesenchymal stem cells, activated stellate cells (as seen in pancreatic cancer), and endothelial cells. Under the influence of tumor derived signals, particularly TGF-β1, these precursor cells undergo phenotypic transformation into CAFs [4].

In a study conducted by Carmen García-Pravia et al., which included 54 cases of pancreatic ductal adenocarcinoma (PDAC) and 23 cases of chronic pancreatitis (CP), qRT-PCR and immunohistochemistry were employed to evaluate COL11A1 expression. The COL11A1 demonstrated a sensitivity of 92% and a specificity of 83% for identifying CAFs [30]. These findings underscore the reliability of COL11A1 as a stromal marker distinguishing tumor-associated fibroblasts from those in benign inflammatory conditions.

Moreover, COL11A1 expression is higher in primary tumors than in metastatic lesions and has been implicated in the activation of the NF-κB pathway, which facilitates the transformation of normal fibroblasts into CAFs. These activated fibroblasts exert pro-inflammatory and immunomodulatory effects through IL-6, promoting an immunosuppressive microenvironment via recruitment of regulatory T cells (Tregs) and M2-polarized macrophages. Additionally, COL11A1 induces excessive collagen deposition, resulting in a dense stromal matrix that hinders T-cell and dendritic cell infiltration, as well as the diffusion of chemokines and cytokines essential for antitumor immune responses [4,31].

In conclusion, COL11A1 plays a pivotal role in shaping the tumor microenvironment by promoting fibroblast activation, stromal remodeling, and immune suppression. These findings suggest that targeting COL11A1 could enhance therapeutic precision and potentially limit tumor progression.

### 5.2. COL11A1-Signaling Pathways in Breast Cancer

COL11A1 overexpression has been associated with distinct mechanisms of therapeutic resistance in breast cancer. The specific composition of the breast stroma, together with tumor microenvironmental heterogeneity and immunological variation among molecular subtypes, may influence the signaling pathways activated by COL11A1 [2,5,8].

To date, three principal signaling pathways have been described through which COL11A1 promotes tumor cell proliferation, invasion, metastasis, and the expansion of the inflammatory tumor infiltrate in breast cancer, see Table 3 [8].

One key pathway involves Oncostatin M (OSM), an inflammatory cytokine released by neutrophils in the tumor stroma. OSM has been shown to bind type XI collagen and activate the STAT signaling cascade, thereby stimulating proliferation and migration in T47D breast cancer cells [4,8,32].

Regulation of COL11A1 expression also occurs at the post-transcriptional level via microRNAs such as miR-139-5p and let-7b. miR-139-5p suppresses COL11A1 expression, leading to reduced tumor cell proliferation. let-7b, upregulated by CDX2 transcriptional activity, downregulates COL11A1 and consequently inhibits tumor cell proliferation, invasion, and migration, see Figure 2 [33,34,35].

These molecular insights reveal potential therapeutic targets that may improve treatment efficacy and help overcome resistance in breast cancer. Understanding COL11A1-mediated signaling pathways is therefore essential for the development of novel, mechanism-based treatment strategies.

### 5.3. The Effect of COL11A1 on the Immune System

Immunotherapy represents a promising treatment option for breast cancer; however, clinical responses remain highly heterogeneous and difficult to predict. The role of COL11A1 in stromal remodeling and its potential interaction with the immune microenvironment have been highlighted in several preclinical and exploratory studies. Although these findings suggest possible implications for immunotherapy, clinical validation in prospective cohorts is still required [2,8].

In a study conducted by Qi Luo et al., increased COL11A1 expression was shown to upregulate immune checkpoints such as PD-1, PD-L1, and CTLA-4, thereby inhibiting T-cell activity and promoting immune evasion [2]. These immune checkpoints represent major therapeutic targets in breast cancer, and their modulation by COL11A1 underscores a potential mechanistic link between the tumor stroma and immune suppression [8].

Furthermore, COL11A1 expression was associated with altered immune cell infiltration, characterized by low levels of B and CD8^+^ T cells and high levels of CD4^+^ T cells [5].

Wenjie Shi et al. further confirmed a correlation between COL11A1 and additional immune checkpoints, including CD276, TIGIT, and ENTPD1, suggesting that elevated COL11A1 expression promotes resistance to immune effector cell activity. Interestingly, patients responding to immunotherapy also displayed higher COL11A1 levels; however, these results remain inconclusive and require further investigation [5].

Studies have also demonstrated that COL11A1 expression differs between “immunologically hot” and “immunologically cold” tumor subtypes [36].

COL11A1 is overexpressed in cold subtypes, characterized by poor lymphocytic infiltration and low immunogenicity, which often results in reduced responsiveness to immune checkpoint inhibitors [37].

### 5.4. Resistance to Hormone Therapy

Endocrine therapy was first introduced for the treatment of breast cancer in 1940 [38]. There are two main estrogen receptor isoforms, ERα and ERβ. ERα is predominantly expressed in the mammary gland, uterus, ovary, bone, testis, and epididymis, whereas ERβ is found mainly in the prostate, urinary bladder, and granulosa cells of the ovary [39]. These receptors function as ligand activated transcription factors that modulate gene expression upon binding estrogen.

Activation of ERα promotes abnormal cell proliferation in breast and uterine tissues, whereas ERβ exerts an antiproliferative effect. Therefore, patients with estrogen-dependent breast cancer are expected to benefit from ERα antagonists or ERβ agonists [38,39].

However, in clinical practice, the situation is more complex. Early-stage ductal carcinomas generally exhibit high ERα and low ERβ expression, profiles compatible with ERα antagonist or ERβ agonist therapies, whereas high-grade ductal carcinomas often lack ER expression. Similarly, early stage lobular carcinomas show high levels of both ERα and ERβ, while high-grade forms express only ERα, favoring ERα antagonist therapy [40].

Resistance to endocrine therapy remains a major clinical challenge and may arise through genetic or epigenetic mechanisms. This can be intrinsic (absence of response from the outset) or acquired (initial response followed by resistance during treatment, relapse, or metastasis). Approximately 25% of patients with early-stage ER-positive breast cancer develop endocrine resistance [41].

One key mechanism involves mutations in the ESR1 gene, which encodes ERα [42]. Such mutations confer resistance to aromatase inhibitors and tamoxifen, a selective estrogen receptor modulator (SERM) that has been used for over three decades. These mutations occur in about 1% of primary tumors and 20–24% of relapsed cases, underscoring the need for therapies that directly target mutant ESR1 [43].

Epigenetic mechanisms also contribute to endocrine resistance. According to Liliana García Martínez et al., these include dysregulation of ERα co-regulators, ligand-independent activation by growth factors, ERα downregulation, mitogenic signaling activation, transcription factor alterations, and chromatin remodeling [44]. Mutations affecting ERα coactivators and corepressors (e.g., NCoR1, NF1) disrupt regulatory balance, promoting uncontrolled proliferation and therapeutic resistance [45]. 

Furthermore, Lloyd et al. demonstrated that alterations in CDK4/6 can lead to endocrine resistance in HR-positive, HER2-negative breast cancers [46]. Beyond these classical mechanisms, recent studies have identified components of the extracellular matrix (ECM), particularly COL11A1, as potential modulators of endocrine resistance [47]. 

Chengxiao Fu et al. conducted in vitro studies using MCF-7 and T47D breast cancer cell lines overexpressing COL11A1 and observed resistance to 4-hydroxy-tamoxifen (4-OHT) compared with parental cells. In these resistant cells, ERα levels were increased, suggesting a potential feedback mechanism induced by COL11A1 overexpression [9].

In conclusion, resistance to endocrine therapy in breast cancer results from a multifactorial interplay of genetic, epigenetic, and microenvironmental factors. While ESR1 mutations and ERα co-regulator dysregulation are well established drivers, emerging evidence indicates that extracellular matrix components, particularly COL11A1, may also contribute to reduced tamoxifen sensitivity and endocrine resistance.

### 5.5. Chemotherapy Resistance

Chemotherapy remains a cornerstone of breast cancer treatment; however, the emergence of chemoresistance poses significant clinical challenges [48].

Therapeutic response is typically assessed through physical examination, imaging, and histopathological evaluation, the latter offering the highest predictive value for disease-free survival (DFS) and overall survival (OS). The Miller and Payne classification is commonly used to grade histopathologic response to treatment [49].

According to Muhammad Faruk et al., key mechanisms underlying chemotherapy resistance include drug inactivation, alterations of drug targets, overexpression of ABC transporters, epithelial-to-mesenchymal transition (EMT), apoptotic dysregulation, and the persistence of cancer stem cells [48].

Recent studies have shown that COL11A1 expression is closely associated with the development of chemotherapy resistance and with increased expression following treatment. Initially, COL11A1 overexpression was linked to resistance to cisplatin in ovarian cancer. In a study by Yi-Hui Wu et al., including 60 patients with epithelial ovarian carcinoma, high COL11A1 expression correlated with older patient age and advanced tumor stage [49].

Subsequent research demonstrated that COL11A1 stimulates the synthesis of inhibitor of apoptosis proteins (IAPs) such as XIAP, BIRC2, and BIRC3, alters tumor cell metabolism, and upregulates TWIST1 expression, collectively contributing to resistance against cisplatin and paclitaxel therapies [50,51].

Among these, one of the most studied genes is MDR1, which encodes P-glycoprotein, a membrane efflux pump that expels cytotoxic agents from the cell. Under physiological conditions, this protein facilitates the excretion of xenobiotics; however, MDR1 overexpression markedly enhances drug efflux, resulting in decreased intracellular drug accumulation and subsequent resistance [52].

While molecular mechanisms such as COL11A1 upregulation and MDR1-mediated drug efflux play central roles in chemoresistance, clinical and tumor related variables including: age, cancer subtype, and disease stage, also significantly influence therapeutic outcomes [53].

Evidence indicates that older patients are more prone to develop resistance, and among molecular subtypes, triple-negative breast cancer (TNBC) exhibits the highest chemosensitivity [14].

In conclusion, chemotherapy resistance in breast cancer arises from a multifactorial interplay between molecular, cellular, and clinical determinants. While traditional mechanisms such as drug efflux, target modification, and apoptotic dysregulation are well established, emerging evidence highlights the contribution of extracellular matrix components, particularly COL11A1, in modulating tumor metabolism, apoptosis, and epithelial–mesenchymal transition, thereby promoting resistance and disease progression.

## 6. Discussion

### 6.1. Biomarker Development

According to the latest global cancer statistics, breast cancer ranks fourth in cancer-related mortality worldwide. Its incidence has now surpassed that of lung cancer, making it the most frequently diagnosed malignancy among women. This trend underscores its major public health impact [1].

A persistent concern is that many cases are still diagnosed at advanced stages, where the benefit of surgery is limited by the presence of distant metastases. Consequently, early detection remains one of the greatest challenges in breast cancer management. Another major challenge is the development of resistance to conventional therapies, emphasizing the need to identify new prognostic markers and therapeutic targets [2,5,8].

Immunotherapy has emerged as a promising alternative for several solid tumors, leading to increasing interest in its application to breast cancer [2].

Wenjie Shi et al. reported that COL11A1 expression positively correlates with immune checkpoint molecules such as CD276 (B7-H3), TIGIT, and ENTPD1 (CD39), with strong statistical significance (*p* < 0.001). COL11A1 expression also appeared higher in patients responding to anti–PD-1 immunotherapy. However, this association did not reach statistical significance (*p* ≈ 0.33) [5].

Although COL11A1 is not yet recognized as a clinically validated biomarker for resistance to immune checkpoint inhibitors (anti–PD-1/PD-L1) in breast cancer, it remains of scientific interest. Several studies suggest that it may contribute to tumor microenvironment remodeling and immunosuppression [28].

### 6.2. Challenges of Clinical Translation

Since COL11A1 expression differs between immunologically “hot” and “cold” tumors, immunohistochemical analysis could help identify patients with “cold” tumors who may benefit from combination therapies rather than immunotherapy alone [5,54].

Currently, there is no standardized universal score for COL11A1 as there is for ER/PR, HER2, or PD-L1. However, principles validated in other contexts can be adopted to establish a rigorous and reproducible methodology [55,56,57].

First, scoring should be performed only in the stromal compartment, avoiding tumor epithelial and immune cells. A scale similar to the H score or Allred scores can be used. The intensity of staining in fibroblasts will be evaluated: 0 = negative, 1 = weak, 2 = moderate, 3 = intense as well as the percentage of positive stromal fibroblasts, between 0–100%. Finally, the H-score can be calculated using the formula (% fibroblasts 1+) × 1 + (% fibroblasts 2+) × 2 + (% fibroblasts 3+) × 3.

Validation should be conducted on clinical cohorts, and the immunohistochemical result should be correlated with COL11A1 mRNA expression [2,8,27].

According to studies, LY2157299 is a small molecule that is capable of inhibiting the TβR1 (Transforming Growth Factor Beta Receptor 1) receptor, involved in the activation of TGF-β (Transforming Growth Factor Beta). This is a cytokine with a role in cell proliferation, differentiation and migration, which also stimulates the expression of the COL11A1 gene. This molecule has passed phase I of studies, entering phase II of clinical trials, for patients with ovarian cancer [4,8,58].

Establishing a standardized COL11A1 scoring method could not only improve patient stratification but also guide the development of targeted therapies.

### 6.3. Limitations

Despite these promising insights, significant knowledge gaps remain. The precise structural organization of COL11A1 in cancer is not yet fully understood. The identity of the two α-chains that assemble with COL11A1, as well as the proteins interacting with its N- and C-terminal domains, remain unknown [4].

Moreover, evidence is still lacking on how COL11A1 modulates the biophysical properties of the tumor extracellular matrix and influences the migration of tumor and immune cells [59].

Although our understanding of its roles in chemoresistance has expanded, its contributions to cancer stemness, tumor dormancy, inflammation, and recurrence are not fully elucidated.

Furthermore, there is limited information regarding the functions of the 28 different collagen subtypes. These subtypes may exert overlapping or compensatory effects, thus diminishing the therapeutic role of COL11A1 inhibition [4].

Another limitation is the absence of large, well powered clinical studies assessing the frequency of COL11A1 expression across breast cancer subtypes or defining its relationship with resistance to endocrine and chemotherapeutic therapies.

Most available evidence derives from retrospective transcriptomic analyses or small immunohistochemical cohorts, which lack the statistical power to establish COL11A1 as an independent prognostic or predictive biomarker [10].

Moreover, many mechanistic insights are extrapolated from ovarian, pancreatic, or lung cancers, raising questions about their applicability to breast cancer biology. At present, there are no validated therapeutic strategies that directly target COL11A1 or its signaling pathways, and whether it acts as a functional driver or merely a stromal bystander remains uncertain [30,60].

Clinical evidence linking COL11A1 expression to therapy resistance is currently limited, particularly for breast cancers. A significant portion of the reported associations, particularly those related to chemotherapy resistance, come from studies in other tumor types (e.g., ovarian cancer) or from preclinical analyses. Extrapolation of these data to breast cancers is warranted only with caution, given the biological differences between tissues, the variability of the tumor microenvironment, and the divergences in therapeutic regimens.

Consequently, conclusions regarding the prognostic or predictive role of COL11A1 in breast cancer should be considered preliminary until validated in prospective, standardized clinical cohorts.

### 6.4. Future Research Directions

Future research should include large, well characterized clinical cohorts stratified by molecular subtype, integrate multiomics profiling of stromal and immune components, and correlate these data with therapeutic outcomes. Such an approach may clarify whether COL11A1 functions merely as a stromal marker of desmoplasia. It may also determine whether COL11A1 acts as an active determinant of immunotherapy sensitivity. This could open avenues for its validation as a predictive biomarker or therapeutic target [7].

The COL11A1-rich desmoplastic stroma forms both a physical and biochemical barrier that limits immune cell infiltration and therapeutic penetration [61,62].

Notably, inhibition of the IL-6/STAT3 pathway activated downstream of COL11A1 has been shown to reduce immunotherapy resistance and enhance response to anti–PD-1/PD-L1 treatment. This effect, particularly in fibrotic microenvironments, is characterized by high COL11A1, low PD-L1, and low CD8 expression [63].

Building upon these findings, several combination strategies have been proposed to counteract COL11A1-driven immunosuppression. For instance, IL-6R inhibitors (e.g., Tocilizumab) combined with anti–PD-1 agents have demonstrated synergistic immune activation in desmoplastic tumor models [64].

Similarly, JAK1/2 inhibitors such as Ruxolitinib, by disrupting IL-6/STAT3 signaling downstream of COL11A1, may further augment antitumor immune responses [63,65].

In summary, increasing evidence supports a central role for COL11A1 in stromal remodeling, immune modulation, and therapy resistance. However, its precise mechanistic functions in breast cancer remain to be fully elucidated. Further research is required to determine its value as a predictive biomarker and its potential as a therapeutic target in personalized oncology.

## 7. Conclusions

Breast cancer requires a multidisciplinary treatment approach, including surgery, chemotherapy, radiotherapy, hormonal therapy, and targeted therapy. Nevertheless, therapeutic resistance remains a major clinical challenge, emphasizing the urgent need for novel predictive biomarkers and therapeutic targets [48].

Chemotherapy remains an essential component of breast cancer management. Therefore, continuous monitoring of treatment response is crucial to prevent or identify resistance at an early stage [48,66].

Similarly, hormonal therapy continues to play a pivotal role in hormone-dependent breast cancer, but the emergence of resistance highlights the necessity for new therapeutic options in these patients [9].

Although significant progress has been made in developing strategies to overcome resistance to chemotherapy and endocrine therapy, immunotherapy has emerged as a valuable alternative, particularly for patients who do not benefit from conventional treatments [2,9,45,48].

However, only a subset of breast cancer patients respond to immunotherapy, underscoring the importance of identifying reliable predictive biomarkers to guide patient selection. Historically, the limited ability to identify tumor-specific immune targets has slowed the clinical implementation of immunotherapy in breast cancer [67,68].

Recent advances have demonstrated that immune checkpoint inhibitors, such as PD-1 and PD-L1 antagonists, can improve outcomes in selected patients with metastatic breast cancer [2,8,69].

Within this context, COL11A1, a fibrillar collagen overexpressed in pathological conditions, has attracted attention due to its association with tumor progression, therapeutic resistance, and prognosis across multiple cancer types, including breast cancer [4,8].

Secreted primarily by cancer-associated fibroblasts (CAFs), COL11A1 may serve as a novel stromal biomarker and a potential therapeutic target within the tumor microenvironment [70,71].

Although differences in disease-free survival (DFS) and overall survival (OS) have been observed in patients with COL11A1 overexpression, these associations have not reached statistical significance, likely due to the limited size of current study cohorts [2,8].

Nonetheless, COL11A1 offers several advantages as a biomarker: it demonstrates high sensitivity and specificity, reflects tumor stromal activity, and provides information independent of traditional epithelial markers such as ER, PR, HER2, or PD-L1 [2,72].

Therefore, while COL11A1 holds significant promise as both a diagnostic and therapeutic target, it cannot yet be placed at the same level as clinically validated biomarkers. Large scale, prospective, and breast cancer-specific studies are urgently required to determine its true incidence, prognostic value, and predictive utility in guiding treatment strategies.

Only through rigorous clinical validation can COL11A1 transition from an exploratory biomarker to a clinically actionable tool in breast cancer management.

In conclusion, COL11A1 represents a promising direction in the search for novel biomarkers that integrate tumor biology, stromal remodeling, and immune interactions.

Future research should aim to confirm its role in prognosis and treatment response, paving the way for its potential inclusion in the routine immunohistochemical profile of breast cancer and contributing to the development of more personalized and effective therapeutic strategies.

## Figures and Tables

**Figure 1 biomedicines-13-02905-f001:**
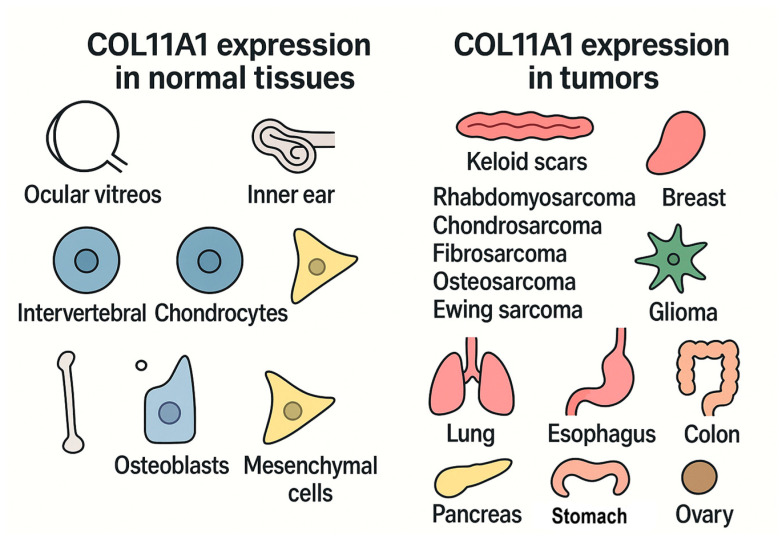
COL11A1 normal and tumor expression.

**Figure 2 biomedicines-13-02905-f002:**
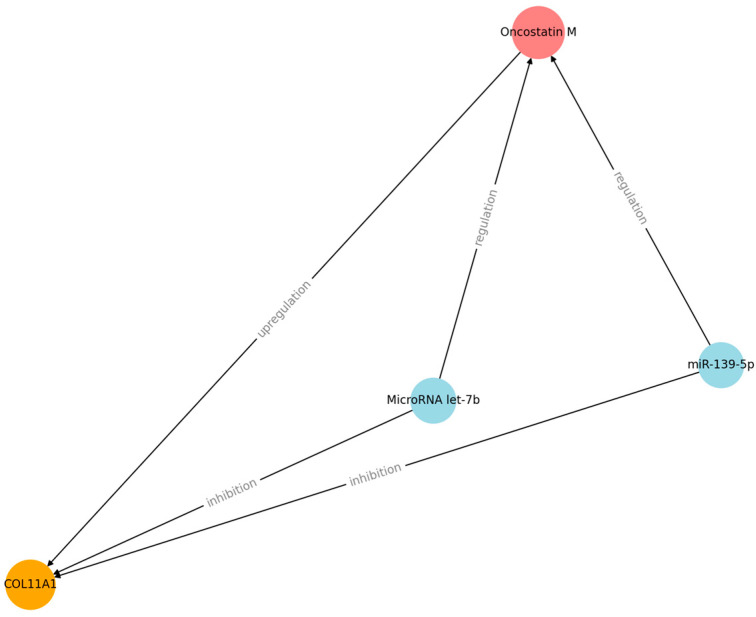
COL11A1-signaling pathways in breast cancer.

**Table 1 biomedicines-13-02905-t001:** General characteristics of the relevant studies.

No.	Reference	Cohort	Method	Key Outcome
1	Luo et al., 2022 [2]	Breast samples	Bioinformatics + IHC	High COL11A1 associated with shorter OS and aggressive phenotypes
2	Shi et al., 2022 [5]	Six breast cancer datasets	Machine learning on transcriptomic data + IHC	Correlates with poor prognosis and altered immune infiltrate
3	Freire et al., 2023 [6]	Small clinical biopsies (including breast)	IHC vs. histology	COL11A1 may predict tumor infiltration; potential clinical utility
4	Wang et al., 2025 [7]	Samples breast cancer tissue, benign breast tumors, normal breast tissue and lymph nodes with metastases	IHC and real-time reverse transcription PCR (RT-PCR)	COL11A1 mRNA is significantly higher in primary breast cancer tissues than in adjacent normal tissue
5	Wu et al., 2022 [8]	Previous studies	Analyze and synthesize the results	Increased expression is frequently associated with aggressive tumors
6	Fu et al., 2024 [9]	MCF-7/T47D cells and MCF-7/COL11A1 and T47D/COL11A1 cells	IHC + in vivo functional assays	High COL11A1 linked to poor tamoxifen response; in vivo resistance confirmed
7	Jia et al., 2016 [10]	Data from The Cancer Genome Atlas (TCGA) for 13 types of primary carcinoma	Transcriptomic, functional analysis and IHC validation.	COL11A1 has been identified as a specific biomarker for activated CAFs in multiple cancer types.

**Table 2 biomedicines-13-02905-t002:** Classical immunohistochemical markers and COL11A1.

**Implications**	ER/PR/HER2/Ki-67	COL11A1
**Application**	Standard	Experimental
**Marker type**	Tumor cell	Peritumoral stroma
**Diagnostic**	Molecular subtype, classification	Confirmation of invasion
**Prognostic value**	Yes	Yes
**Targeted therapy**	Hormonal therapy, anti-HER2 therapy	Future treatment option

**Table 3 biomedicines-13-02905-t003:** The mechanism of COL11A1 regulation in breast cancer.

Regulator	Effect
miR-139-5p	Increases cell proliferation and inhibits apoptosis
MicroRNA let-7b	Increases cell proliferation, migration, invasion, and metastasis
Oncostatin M	Increases inflammation and metastasis

## Data Availability

No new data were created or analyzed in this study. Data sharing is not applicable to this article.

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
