# Peer review of "Stromal COL11A1: Mechanisms of Stroma-Driven Multidrug Resistance in Breast Cancer and Biomarker Potential"

_biomedicines, 2025, doi:10.3390/biomedicines13122905_

Round 1

Reviewer 1 Report (Previous Reviewer 1)

Comments and Suggestions for Authors

The manuscript can be accepted in current form

Author Response

Dear Reviewer,

Thank you very much for your positive evaluation of our work. We appreciate your time and effort in reviewing the manuscript and are glad to hear that it is suitable for acceptance in its current form.

Best regards,

Andreea Onofrei

Reviewer 2 Report (Previous Reviewer 2)

Comments and Suggestions for Authors

Dear authors,

Thanks for your submission. Just two minor issues should be pay attention.
1.Some misspelling in Figure.1, such as stomac, brast, ovar. Please correct them.
2.Sections such as epidemiology and histological and molecular classification provide extensive background information that is only tangentially related to COL11A1. These could be substantially condensed to improve focus and readability.

Thank you.

Author Response

 Dear Reviewer,

Best regards,

Daniela Mihalache

Reviewer 3 Report (New Reviewer)

Comments and Suggestions for Authors

This review systematically explores the role of COL11A1 in breast cancer drug resistance, demonstrating certain scientific value and clinical significance. However, significant improvements are needed in content focus and structural optimization.

  1. The title emphasizes "immunohistochemical biomarker," but the review's content actually covers a broader scope (including molecular mechanisms, signaling pathways, multidrug resistance, etc.), which may limit readers' expectations regarding the article's comprehensiveness. Furthermore, the title needs to highlight "multidrug resistance." It is recommended to revise the title, for example, to: Stromal COL11A1: Mechanisms of Stroma-Driven Multidrug Resistance in Breast Cancer and Biomarker Potential

  2. The sections on 3. Breast Cancer Epidemiology and 4. Histological and Molecular Classification are overly lengthy, occupying significant space to introduce basic knowledge of breast cancer. This content is considered general knowledge within the field and lacks strong specific relevance to COL11A1. It is recommended to substantially condense Sections 3 and 4, merging them into a concise introduction. The focus should be on highlighting the current status of treatment resistance in breast cancer, clinical challenges, and the limitations of existing biomarkers to emphasize the unique value proposition of COL11A1 as a stromal marker.

  3. The excessive number of sections suggests a need for structural re-optimization. It is recommended to reorganize into: 1. Introduction, focusing on the current landscape of breast cancer drug resistance to highlight the research value of COL11A1; 2. Biological characteristics of COL11A1 (molecular properties, expression patterns, cellular origins); 3. COL11A1 and mechanisms of therapy resistance (endocrine therapy resistance, chemotherapy resistance, immunotherapy resistance); 4. Clinical translational value (diagnostic value, prognostic significance, therapeutic guidance); 5. Challenges and Future Perspectives; 6. Conclusion.

  4. Breast cancer-specific clinical evidence is limited. Some inferences are based on other cancer types; for example, the association between COL11A1 expression and chemotherapy resistance is primarily based on studies in ovarian cancer. A critical analysis discussing the limitations of the current evidence is necessary.

Author Response

Dear Reviewer,

Best regards,

Daniela Mihalache

Round 2

Reviewer 3 Report (New Reviewer)

Comments and Suggestions for Authors The authors' current response fully addresses all the comments raised by the reviewer, with a sincere attitude and strong relevance in the replies. However, the response to Comment 2 fails to meet the reviewer's core requirements. The streamlined section "Epidemiology and Classification of Breast Cancer" has unreasonable data selection, deviates from the core goal of "serving the research value of COL11A1", and suffers from the issue of "one sentence per paragraph" with fragmented sentence structures. It is recommended to retain core data to support the "severity of drug resistance", delete redundant content such as "subdivided risk factors and proportions of histological types", and revise the section following the logical framework below: Breast cancer is the most common malignancy among women worldwide, with over 2.3 million new cases diagnosed annually since 2020. Approximately 30% of patients develop therapeutic resistance during the disease course, which has become the leading cause of death [6,8]. Although molecular subtyping (Luminal A/B, HER2-enriched, basal-like, etc.) has advanced personalized treatment, the challenge of drug resistance remains unresolved — for instance, triple-negative breast cancer (TNBC), which lacks well-defined therapeutic targets, has a chemotherapy resistance rate exceeding 50%; about 60% of HER2-positive breast cancer patients develop secondary resistance within 1-2 years of targeted therapy [25,26]. Existing biomarkers associated with drug resistance mostly focus on tumor cells themselves (e.g., ER/PR expression, HER2 amplification, BRCA mutations), while overlooking the regulatory role of the tumor microenvironment (TME). As core components of the TME, stromal cells and the extracellular matrix (ECM) they secrete play a pivotal role in drug resistance. However, there are very few studies on stromal biomarkers to date, resulting in a lack of effective predictive and interventional targets in clinical practice [7,19]. From the perspective of subtype characteristics, the drug resistance mechanisms of different breast cancer subtypes are closely linked to the stroma: endocrine resistance in Luminal breast cancer is often accompanied by enhanced stromal fibrosis; "cold tumor" subtypes such as basal-like 1 (BL1) and mesenchymal (M) in TNBC are insensitive to both chemotherapy and immunotherapy due to a thick stromal barrier and poor immune cell infiltration [25,26]. This suggests that targeting stromal components may emerge as a new direction to overcome drug resistance, and the role of COL11A1 — a stroma-specific molecule discovered in recent years — in regulating drug resistance urgently needs to be elucidated.

Author Response

Dear Reviewer,

Best regards, 

Daniela Mihalache 

This manuscript is a resubmission of an earlier submission. The following is a list of the peer review reports and author responses from that submission.

Round 1

Reviewer 1 Report

Comments and Suggestions for Authors

The manuscript provides a comprehensive and relevant review of the current knowledge regarding the role of stromal COL11A1 in resistant breast cancer. The authors discuss the biomarker’s potential as both a diagnostic/prognostic marker and a therapeutic target, particularly in the context of hormone and chemotherapy resistance and emerging immunotherapeutic strategies.

Comments

  1. While the review touches on several mechanisms by which COL11A1 contributes to treatment resistance and tumor progression, further in-depth discussion of unresolved or controversial aspects in the underlying biology would strengthen the manuscript. Please clarify the limitations and gaps in current mechanistic understanding.
  2. The manuscript appropriately notes the lack of standardized assays for COL11A1. I recommend elaborating on the translational pathway, detailing what is needed for clinical adoption of COL11A1 as a biomarker and how it may be integrated into current practice.
  3. The connection between COL11A1 and immunotherapy response is intriguing but remains largely preliminary. Please emphasize the early stage of such findings, the heterogeneity among breast cancer subtypes, and the need for further clinical validation.
  4. As a narrative review, the manuscript lacks original or meta-analytic data. Including a summary table of key clinical studies linking COL11A1 expression to breast cancer outcomes, if available, would increase practical value for readers.
  5. While generally well written, occasional grammatical errors and some lengthy or complex sentences are present. I suggest a careful language edit to improve clarity and readability.

Recommendation

I recommend minor revision, addressing the above points to improve clarity, translational perspective, and overall impact.

Comments on the Quality of English Language
  1. While generally well written, occasional grammatical errors and some lengthy or complex sentences are present. I suggest a careful language edit to improve clarity and readability.

Reviewer 2 Report

Comments and Suggestions for Authors

Dear author,

Thanks for your submission.
This manuscript provides a comprehensive overview of COL11A1 in the context of breast cancer, highlighting its potential as a diagnostic and prognostic biomarker and as a therapeutic target. The topic is clinically relevant, timely, and of potential interest to the readership. However, the paper would benefit from deeper analysis, clearer methodological transparency, and stronger discussion of translational relevance. Here are some comments for you.
1. Usually, we use full name instead of abbreviation at the first showing up which will make readers confusing. (Line 26 IHC)
2. Repetitive abbreviation like TGF-beta. What is Luminal AR? AR= androgen receptor? ER/PR= estrogen receptor/progesterone receptor? These Should be clear. 
3. Several mechanistic insights are extrapolated from ovarian, lung, or pancreatic cancer. These should be clearly separated and critically discussed in terms of relevance to breast cancer.
4. The review presents a large body of evidence but does not adequately compare, contrast, or evaluate the strength and limitations of the cited studies. A more critical synthesis is needed.
5. I suggest to summarize strengths and limitations of COL11A1 compared with established biomarkers (ER, PR, HER2, PD-L1).
6. The literature search methodology is insufficiently detailed. Please specify databases, time range, number of studies identified, and inclusion/exclusion criteria.
7. The section on COL11A1 and classical immunohistochemical markers is too brief and lacks in-depth discussion. A more detailed comparison with established markers and evaluation of its added clinical value would significantly strengthen this part.
8. The conclusion is broad and somewhat repetitive. It should more critically balance the potential of COL11A1 with the current limitations in evidence and clinical applicability.

Looking forward to hearing from you.

Reviewer 3 Report

Comments and Suggestions for Authors

Too many errors needed to be corrected, but several critical notes prevented the manuscript from proceeding to publication.
1. The method was unclearly described as a narrative review, but the method used resembled a systematic review without transparency. This gave the impression of a weak methodology and lacked replicability.

2. The immunotherapy claim was made too quickly and the overclaiming of COL11A1 as an immunotherapy target was immediately positioned despite limited evidence, mostly from preclinical studies. Claims of clinical translation without a strong data base were considered speculative and undermined credibility.

3. The narrative flow in the abstract and introduction was disjointed and poorly focused.  The conclusion also contained irrelevant historical information about immunology. The disorganized flow obscured the main message.

4. There was a lack of emphasis on novelty and research gaps. The manuscript did not clearly indicate its novel contribution to the existing literature. Research gaps (e.g., the frequency of COL11A1 expression in breast cancer or its relationship to therapy resistance) were not emphasized.

5. It was stated that it was used for data collection, which raises questions about academic ethics. AI should not replace the primary literature search process.

Round 2

Reviewer 2 Report

Comments and Suggestions for Authors

Dear author,

Thanks for your resubmission.

Author Response

Dear Reviewer,

Thank you for your time and effort in reviewing our manuscript. We are grateful for your positive evaluation of the manuscript.

Thank you for reconsidering our manuscript.

Sincerely,

Andreea Onofrei

Reviewer 3 Report

Comments and Suggestions for Authors

Thank you to the authors for their improvements to the manuscript. I see significant progress.

However, there is one crucial aspect that requires careful attention: the presentation of the data in Table 1. Currently, the reference format used does not meet scientific publication standards and exhibits significant inaccuracy. I found several issues, such as the use of database names (PMC) and publishers (MDPI) as sources, as well as incomplete citations.

In scientific research, accurate citations are crucial for maintaining transparency and enabling peer verification. The current inaccuracies in Table 1 may cast doubt on the thoroughness of the research process as a whole. I believe the authors can improve this by thoroughly revising Table 1 to ensure each entry has complete, accurate, and verifiable citations.
